# Role of the Innate Immunity Signaling Pathway in the Pathogenesis of Sjögren’s Syndrome

**DOI:** 10.3390/ijms22063090

**Published:** 2021-03-17

**Authors:** Toshimasa Shimizu, Hideki Nakamura, Atsushi Kawakami

**Affiliations:** 1Clinical Research Center, Nagasaki University Hospital, 1-7-1 Sakamoto, Nagasaki City, Nagasaki 852-8501, Japan; 2Department of Immunology and Rheumatology, Unit of Advanced Preventive Medical Sciences, Division of Advanced Preventive Medical Sciences, Nagasaki University Graduate School of Biomedical Sciences, 1-7-1 Sakamoto, Nagasaki City, Nagasaki 852-8501, Japan; nhideki@nagasaki-u.ac.jp (H.N.); atsushik@nagasaki-u.ac.jp (A.K.)

**Keywords:** innate immunity, pattern-recognition receptor, Toll-like receptor, type I interferon, salivary gland epithelial cell

## Abstract

Sjögren’s syndrome (SS) is a systemic autoimmune disease characterized by chronic inflammation of the salivary and lacrimal glands and extra-glandular lesions. Adaptive immune response including T- and B-cell activation contributes to the development of SS. However, its pathogenesis has not yet been elucidated. In addition, several patients with SS present with the type I interferon (IFN) signature, which is the upregulation of the IFN-stimulated genes induced by type I IFN. Thus, innate immune responses including type I IFN activity are associated with SS pathogenesis. Recent studies have revealed the presence of activation pattern recognition receptors (PRRs) including Toll-like receptors, RNA sensor retinoic acid-inducible gene I and melanoma differentiation-associated gene 5, and inflammasomes in infiltrating and epithelial cells of the salivary glands among patients with SS. In addition, the activation of PRRs via the downstream pathway such as the type I IFN signature and nuclear factor kappa B can directly cause organ inflammation, and it is correlated with the activation of adaptive immune responses. Therefore, this study assessed the role of the innate immune signal pathway in the development of inflammation and immune abnormalities in SS.

## 1. Introduction

Sjögren’s syndrome (SS) is an autoimmune disease that mainly causes chronic inflammation of the exocrine glands, resulting in formation of glandular lesions along with ocular and oral dryness and extra-glandular lesions that spread to systemic organs [1]. Polyclonal hypergammaglobulinemia is commonly observed, and several autoantibodies including rheumatoid factor and antinuclear, anti-Ro/SS-A, and anti-La/SS-B antibody are produced [2]. In addition, other autoantibodies against M3 muscarinic acetylcholine receptors (M3R) expressed in salivary gland epithelial cells (SGECs) and those against alpha (α)-fodrin, a type of cytoskeleton protein, may be observed [3,4]. The disease type of SS is generally divided into two categories: primary SS without other autoimmune diseases and secondary SS with other autoimmune diseases such as rheumatoid arthritis and systemic lupus erythematosus (SLE).

An epidemiological study conducted in Japan in 2011 [5] reported that about 68,483 individuals present with SS, with a prevalence rate of 0.05%. The average age of the participants was 60.8 ± 15.2 years. Further, the male:female ratio was 1:17.4, and the peak age at onset was 40–60 years.

Genetic predisposition [6], endocrine abnormalities, including female hormone deficiency [7], and bacterial and viral infections such as Epstein–Barr virus [8] and human T lymphotropic virus 1 [9] are correlated with the development of SS. Moreover, the destruction of some salivary gland tissues can induce inflammation. Therefore, autoantigens such as Ro/SS-A protein, La/SS-B protein, M3R, α-amylase, heat shock 10/60 protein, and α-fodrin flow out owing to cell destruction [10]. The antigen-presenting and SGECs present these autoantigens as antigen peptides to antigen-specific autoreactive T cells, leading to T cell activation [11]. Activated T cells produce cytokines such as interleukin (IL)-2, and further proliferation of T cells promotes the production of inflammatory cytokines such as interferon gamma (IFN-γ), IL-17, and tumor necrosis factor-α. These cytokines act on acinar and ductal epithelial cells and enhance the expression of costimulatory and adhesion molecules such as CD40, CD80, CD86, and ICAM-1 on the cell surface, thereby enhancing inflammatory response. Furthermore, cytotoxic T cells enhanced by these immune responses induce apoptosis in acinar and ductal epithelial cells via Fas ligand/interactions, perforin, and granzymes, leading to advanced glandular tissue destruction, tissue damage, and chronic sialadenitis [12,13,14,15,16]. In addition, cytokines, including IL-6, produced from activated T cells induce the activation of polyclonal B cells, causing autoantibody production and lymphoproliferative pathology.

The histological features of the salivary and lacrimal tissues in patients with SS include atrophy of acinar cells, dilation of ductal epithelial cells, and infiltration of mononuclear cells, mainly CD4+ T and B cells surrounding these cells [17,18,19]. Thus, adaptive immune abnormalities, which are caused by activated T and B cells, play an essential role in tissue disorders and chronic inflammation in SS.

However, the innate immune response, which is a reaction to tissue destruction triggered by microbes, is also activated by multiple signaling cascades, which do not only present antigens but also cause direct inflammation and tissue damage. Moreover, the adaptive immunity further activates cells involved in the innate immunity and induces immune activity and chronic inflammation. Thus, the innate immune response may be significantly involved in not only the initial response in SS pathogenesis but also the development of chronic inflammation.

Currently, there is no radical treatment for SS. The systemic administration of corticosteroids and immunosuppressive agents may be used for severe organ lesions such as those in interstitial lung disease. However, symptomatic treatment is mainly utilized for other extra-glandular or glandular lesions [20]. To develop specific therapies, the cause and pathogenesis of SS, which has different symptoms and can cause systemic lesions, must be identified.

Therefore, this review article evaluated the role of innate immune signaling in SS pathogenesis.

## 2. Innate Immune Response

The innate immunity is a biological defense mechanism that immediately induces inflammation and immune responses after infections. The primary cells involved in the innate immune response include dendritic, NK, and epithelial cells and macrophages [21,22,23]. These cells, referred to as antigen-presenting cells, do not only activate the adaptive immune response but also induce immune responses and inflammation via the signaling pathway.

Innate immunity originates from a pattern recognition receptor (PRR) that recognizes the pathogen-associated molecular pattern (PAMP), which has different molecular structures derived from foreign microorganisms [21,24]. Toll-like receptor (TLR) is the first PRR identified, and it can recognize 10 different PAMP types in humans [24]. TLR1, TLR2, TLR4, TLR5, and TLR6, which are involved in the identification of proteins and fats, are mainly localized on the cell surface. Meanwhile, TLR3, TLR7, TLR8, and TLR9, which are involved in the detection of nucleic acids, are mainly localized within the cell in the endosomes and endoplasmic reticulum (Figure 1) [24].

PRR recognizes danger associated with molecular patterns (DAMPs) in addition to PAMP. Furthermore, it can identify not only environmental factors but also endogenous factors such as autoantigens and peptides, which trigger the innate immune response [25].

PPR-dependent signaling activation is commonly associated with not only inflammation via the nuclear factor kappa B (NF-κB) pathway and mitogen-activated protein kinase (MAPK) pathway but also type I IFN production (Figure 1). The genome-wide association study revealed that genes associated with susceptibility to SS (such as interferon regulatory factor 5 (*IRF5*), signal transducer and activator of transcription 4 (*STAT4*), and 2′-5′ oligoadenylate synthetase 1 (*OAS1*)) encode transcription factors correlated with type 1 IFN activity [26,27]. In fact, patients with SS express gene polymorphisms and mRNAs that exhibit excessive type I IFN immune responses [28,29,30,31]. The upregulation of the IFN-stimulated genes induced by type I IFN is referred to as the type I IFN signature [32], which is observed in 50–80% of patients with SS [33]. Recently, the type I IFN signature plays a significant role in immune response imbalance in SS.

Suppressing adaptive immune response-triggering antigen-specific immune responses and innate immune responses that induce inflammation can reduce subsequent chronic inflammation and tissue damage. Therefore, they can be used as a target for radical treatment. Recently, several studies have assessed the importance of innate immune signaling activity in SS pathogenesis.

In addition, SGECs, which are the target cells in the immune response, play an essential role in activating the immune response, including failure of local immune tolerance in SS [34]. The long-term culture system of SGEC strains originating from the lobules of the minor salivary glands was established in vitro, and this is considered a breakthrough in research regarding epithelial cells [35,36].

## 3. Role of Cell Surface TLRs in SS

### 3.1. TLR1, TLR2 and TLR6 in SS

TLR1, TLR2, TLR4, TLR5, and TLR6 recognize proteins and lipids and produce inflammatory cytokines, and they are mainly expressed on the surface of macrophages and epithelial cells [24].

TLR2 forms heterodimers with TLR1 and TLR6 that modify its specificity, thereby facilitating the identification of ligands. In a female nonobese diabetic strain (NOD/Lt) mouse model of SS, the mRNA expression of *TLR1*, *TLR2*, *TLR4*, and myeloid differentiation primary response 88 (*MyD88*) increased in the submandibular glands after disease onset [37]. The expression of these molecules was correlated with a higher lymphocyte infiltration in the submandibular glands, indicating that TLR activation plays a role in recruiting lymphocytes to the salivary glands [37]. In a study using a different mouse model of SS (NOD.B10 mice), female *MyD88* knockout NOD.B10 mice (NOD.B10^MyD88−/−^) did not present with decreased saliva secretion and lower autoantigen production [38]. The administration of TLR4 ligand lipopolysaccharide (LPS) reduced saliva secretion and increased the production of inflammatory cytokines in the submandibular gland tissue in C57BL/6 mice [39].

TLR2 was expressed by minor salivary gland tissues in patients with SS, and this phenomenon was correlated with salivary gland inflammation severity [40]. TLR2 stimulation with peptidoglycan in SS-derived cultured SGECs enhanced the expression of ICAM-1, CD40, and MHC-class I [41]. In addition, TLR2 ligand stimulation in SS-derived cultured SGECs promoted IL-15 secretion in an NF-κB-dependent manner [41]. IL-15 is involved in the proliferation of activated T and B cells and in the maintenance of NK cells [42,43]. One report showed that IL-15 was expressed by acinar and ductal epithelial cells in the salivary glands in SS [44]. TLR2 signaling activity promoted IL-15 production, which indicates that IL-15 can facilitate the survival and proliferation of innate immune system cells such as NK cells and adaptive immune system cells in the salivary glands. The TLR2 expression levels were higher in PBMCs collected from patients with SS than in those obtained from controls, and TLR2 stimulation in SS-derived PMBCs increased IL-17 and IL-23 production [40]. A higher level of IL-17 and activation of Th17 cells that produce IL-17 were observed in the salivary glands and peripheral blood, indicating that TLR2 signaling promotes the differentiation of T cells into Th17 cells and enhances IL-17 production in patients with SS [45].

### 3.2. TLR4 in SS

Moreover, TLR4 was expressed by infiltrating mononuclear cells and acinar and ductal epithelial cells in the salivary glands in patients with SS, and this phenomenon was correlated with salivary gland inflammation [45,46]. Stimulation with the TLR4 ligand LPS enhanced the expression of costimulatory and adhesion factors (ICAM-1, CD40, and MHC-class I) by SS-derived cultured SGECs [41]. In addition, LPS stimulation upregulated TLR4 and promoted the secretion of inflammatory cytokines and chemokines IL-6, IL-12, CCL5, GM-CSF, and MCP-1 in the A253 salivary gland cell line [47]. Furthermore, our study showed the stimulation of SS-derived cultured SGECs with peptidoglycan and LPS induced the phosphorylation of MAPK family, including extracellular signal-related kinase, c-Jun N-terminal kinase, and p38 [46]. Results showed that TLR ligand stimulation promotes MAPK pathway activity in the salivary glands in SS. Another study revealed that mucin in the saliva activates TLR4, and it is involved in chronic inflammation. Hence, this glycoprotein can be a candidate ligand for TLR4-dependent signaling [48].

### 3.3. TLR5 in SS

The stimulation of the flagellar filament structural protein FliC, a TLR5 ligand, caused salivary gland inflammation and increased serum inflammatory cytokine levels and IgG and anti-Ro/SS-A antibody levels in C57BL/6 mice [49]. Therefore, the TLR5 signaling activity can promote salivary gland inflammation and autoantibody production. However, based on a previous study, the TLR5 expression in PBMCs decreased in individuals with SS compared with healthy controls [50]. However, data about this topic are extremely limited; thus, further studies must be performed.

## 4. Role of Endosomal TLRs in SS

### 4.1. TLR3, TLR7–9 in SS

TLR3 and TLR7–9 are localized to the endoplasmic reticulum and within endosomes, and they recognize nucleic acids and promote inflammatory cytokine signaling and type I IFN production signaling activity [24]. Moreover, they are primarily expressed by innate immune cells such as plasmacytoid dendritic cells (pDCs) and epithelial cells. Although TLR3 mainly recognizes dsRNA produced by viruses, it can also detect endogenous RNA released by necrotic cells [25,51,52].

In NOD/Lt mice carrying two genome regions, which are involved in the development of SS (*Idd* susceptibility loci), a microarray analysis revealed an increased expression of TLR3 and the downstream signaling molecule tumor necrosis factor receptor-associated factor 6 (TRAF6) and interferon regulatory factor (IRF) 3 [53,54]. Female New Zealand Black/WF1 mice with systemic lupus erythematosus (SLE) and SS-like lesions expressed TLR3 in the salivary gland, particularly in duct epithelial and acinar cells. The administration of polyinosinic:polycytidylic acid (poly I:C), a TLR3 ligand, reduced saliva secretion and increased sialadenitis [55,56]. The stimulation of TLR3 also enhanced the expression of inflammatory cytokines and chemokines (IFNβ, IL-6, IL-1β, and CCL5) and IFN-stimulated genes (ISGs) in the salivary glands [55,56]. In addition, poly I:C stimulation did not reduce salivary function in IFN-α/β receptor^−/−^ or IL-6^−/−^ mice, indicating that type I IFN and IL-6, which are induced in the TLR3-dependent signaling pathways via poly I:C stimulation, can directly impair saliva secretion [57]. TLR3 was expressed by cultured SGECs collected from patients with SS, and the poly I:C stimulation of these SGECs increased the expression of costimulatory factors and adhesion molecules (ICAM-1, CD40, and MHC-class I) [41].

As mentioned in the previous text, activated T cell-mediated apoptosis resulted in glandular tissue damage in patients with SS [15,16]. In addition, TLR3 signaling activity might play a role in the apoptosis of SGECs in patients with SS [58,59,60]. Poly I:C stimulation induced apoptosis via the upregulation of receptor-interacting protein kinase 3, phosphorylated fas-associated death domain, and caspase-8 [59,60]. In addition to apoptosis, TLR3 ligand stimulation of SGECs enhanced the expression of Ro/SS-A and La/SS-B, which are SS autoantigens [61]. These findings indicated that TLR3-dependent signaling caused tissue damage by inducing the apoptosis of SGECs and provoked an immune response by releasing autoantigens that elicit immune activity. However, our study showed that SGEC apoptosis was induced by TLR3 stimulation in vitro, and the expression of these apoptotic molecules is suppressed in the salivary glands in vivo in SS. In addition, epidermal growth factor (EGF), which is produced by the salivary glands, induced the expression of anti-apoptotic proteins such as heme oxygenase-2 and heat shock protein-27. This phenomenon indicates that EGF regulates apoptosis in the salivary glands in SS [60].

Next, we focused on the expression of TLR7–9 in patients with SS. TLR7 and TLR8 were expressed in cultured SGECs, salivary gland tissues, and peripheral mononuclear cells (PBMCs) collected from patients with SS [50,62,63,64,65,66].

When we assessed TLR7–9 expression and function in the salivary glands in SS, the enhancement of TLR7 expression in the minor salivary glands was particularly evident in ductal epithelial cells, pDCs, and B cells in SS [66]. In addition, molecules downstream of the TLR7-dependent signaling (MyD88, TRAF6, and IRF7) were expressed by ductal epithelial cells and pDCs. Type I IFN expression was enhanced in ductal epithelial cells and pDCs. However, there was no enhancement of TLR9 expression. In addition, the stimulation of SS-derived cultured SGECs with the TLR7 ligands loxoribine and IFNβ activated TRAF6 and IRF7, which are downstream signaling molecules, and enhanced the expression of Ro52/TRIM21 and MHC-class I. Hence, type I IFN production occurs via the TLR7 signaling activity not just in mononuclear cells that infiltrate the salivary glands, including pDCs, but also in the ductal epithelium itself. Furthermore, the TLR7 signaling activity in SGECs may promote autoantigen production and antigen presentation via MHC-class I.

Another in vitro study regarding the MAPK and Janus kinase (JAK)/STAT pathways showed increased activation of NF-κB and STAT3 in response to TLR7, TLR9 stimulation particularly in peripheral B cells in SS patients with anti-Ro/SS-A antibody and those without extra-glandular lesions [67]. In addition, the increased activation of these pathways is correlated with the type I IFN signature. Indeed, the expression of STAT3, JAK1, and JAK2 increased in the salivary glands of patients with SS [40,68]. Moreover, the inhibition of JAK suppressed the expression of IFN-related genes in SS-derived cultured SGECs. JAK inhibitor-treated NOD mice also showed increased salivary flow rates and reduced lymphocytic infiltration in salivary glands [69]. These findings suggested that the JAK/STAT pathway activated upon type I IFN signaling could contribute to SS pathogenesis.

Some reports have shown increased TLR9 expression in the minor salivary glands, parotid tissue, and PBMCs in patients with SS [64,65]. However, these findings are contradicted by another study showing a reduced TLR9 expression in the salivary glands, PBMCs, and monocytes in patients with SS [50,63,66,70]. Furthermore, in NOD/Lt mice, stimulation with a TLR9 agonist induced an anti-inflammatory response and increased saliva secretion [70].

Unc93B1 is a protein responsible for trafficking TLR7 and TLR9 to the endosome where TLR7 and TLR9 recognize nucleic acids. In addition, Unc93B1 is known as one of the balance control mechanisms between TLR7 and TLR9 reactions. The disruption of this control activates TLR7 signaling and leads to inflammation [71]. In fact, previous studies showed that TLR7 ligand stimulation in human PMBCs suppressed the expression of *TLR9* mRNA [63]. TLR9 activity plays a protective role against SLE [72]. Thus, it may also be suppressed, and TLR7 activity is favored by a Unc93B1-mediated control mechanism during SS pathogenesis.

### 4.2. TLR7, TLR9, and Sex Differences in SS

As mentioned in the previous text, the proportion of female patients with SS was higher than that of male patients [5]. The association between female hormones and the development of SS has also been observed in estrogen-deficient mice [73].

In addition, the gene dosage effect associated with X-chromosome genes is involved in increased TLR7 signaling in SS. In women, either the maternal or paternal X chromosome is randomly silenced in a process called X-chromosome inactivation (XCI) [74]. However, this epigenetic change is not observed in 100% of cases. A study using gene expression datasets showed an increased expression of 58 X-chromosome genes in the salivary glands in patients with SS, in which 22 genes had escaped XCI [75]. Notably, the gene encoding TLR7 is on the X chromosome, and it escapes XCI in immune cells, which may lead to increased TLR7 copy numbers and TLR7 hypersensitivity in women [76]. Therefore, enhanced TLR7 responses in SS may be caused by improper gene silencing.

TLR7 expression may be dependent on sex hormone levels. Androgens, particularly in the blood and tissue, can reduce the expression of TLR7 and TLR9 in pDCs. Further, androgens protect epithelial cells against apoptosis and reduce autoantigen release, resulting in decreased TLR expression. Thus, lower sex hormone levels and localized loss of androgen production after menopause may cause increased TLR expression by pDCs. This phenomenon then promotes pDC activation and loss of immune tolerance in female patients [77].

Therefore, sex differences and abnormal TLR7–9 function may play a contributory role in SS pathogenesis.

## 5. Role of Cytoplasmic RNA and DNA Sensors in SS

Nucleic acid recognizing sensors that induce type I IFN production are localized in the cell cytoplasm. These include RNA sensor retinoic acid-inducible gene I (RIG-I) and melanoma differentiation-associated gene 5 and DNA sensor cyclic GMP-AMP synthase (cGAS) [78,79].

A previous study used IFN score to stratify patients as type I IFN signature-positive (IFN score of ≥10) and type I IFN signature-negative (IFN score of <10). Results showed that the expression of TLR7- and TLR7-dependent downstream signaling molecules MyD88, RSAD2, and IRF7 increased in peripheral pDCs and monocytes in type I IFN signature-positive patients. The expression of RIG-I and MDA5 increased in peripheral pDCs and monocytes in type I IFN signature-positive patients. Moreover, infiltrating cells strongly expressed RIG-I and MDA5 in the salivary glands in patients with SS [63]. Thus, the activation of the RIG-I and MDA5 pathways promoted the production of type I IFN in SS. In addition, triggering of the TLR7 ligand in PBMCs induced the upregulation of RIG-I and MDA5 in vitro. By contrast, the upregulation of cGAS, a stimulator of the interferon gene (STING) pathway, has been detected in patients with SLE, but not in those with SS [80]. Thus, further studies about the cGAS-STING pathway in SS must be conducted.

Collectively, these PRRs in patients with SS produce type I IFN, which then stimulates type I IFN receptors expressed by pDCs and other cells and causes the expression of ISGs. IFN-α can also cause direct cell damage and promote the expression of Ro52/TRIM21 and other autoantigens [81]. Consequently, the immune complex formation between autoantibodies and autoantigens can re-activate TLRs and other PRRs via the FcgRIIa receptor on pDCs and can create an immune activation loop via type I IFN overproduction. As mentioned in the previous text, TLR7 ligand stimulation enhances the mRNA expression of RNA sensors such as *RIG-I* and *MDA5* [63]. Accordingly, TLR7 ligand stimulation can activate other PRRs to synergistically produce the type I IFN signature.

## 6. Role of Endosomal TLRs in Type I IFN-Mediated B Cell Activation

In particular, TLR7 is expressed by B cells [66]. In patients with SS, the TLR7 stimulation of peripheral blood B cells significantly increased the production of cytokines such as IFN-α and IL-6 [82]. The TLR7 and TLR9 ligand stimulation of naïve B cells enhanced differentiation into plasma cells and antibody class switching in patients with SS [83].

In addition, autoantigens and, particularly, nuclear antigens released from apoptotic cells cause dual activation of B cell receptors and TLR signaling, which promotes the disruption of immune tolerance in B cells [84]. For example, the SS autoantigens Ro/SS-A and La/SS-B may form complexes with RNA that are recognized by TLR3, TLR7, and TLR8 expressed by B cells, leading to the activation of these cells [85,86]. In addition, type I IFN specifically induced polyclonal B cell expansion and differentiation via TLR7 signaling in naïve B cells in patients with SS [87]. The expression of not only TLR7 but also ISG increased in serum B cells and in most salivary gland tissues in patients with SS. This phenomenon is correlated with anti-Ro/SS-A and anti-La/SS-B antibody production [88].

Other than the direct effects on B cells, type I IFN-induced signaling can also cause B cell-activating factor (BAFF) production by various cells and facilitate B cell activation. BAFF promotes the survival of immature and mature B cells [89,90]. BAFF-transgenic mice developed autoimmunity, with pathogenic features resembling SS [91]. A previous study showed that the stimulation of SS-derived cultured SGECs with the TLR3 ligand poly I:C and the dsRNA virus reovirus-1 increased BAFF secretion [62]. In addition, BAFF secretion in SS-derived cultured SGECs decreased after pretreatment with chloroquine, which inhibits the activation of TLR3, TLR7, and TLR9. Further, SS-derived cultured SGECs stimulated with IFN-α and IFN-γ produced BAFF [92]. Thus, SGECs from SS are some of the primary producers of BAFF. In fact, the BAFF expression levels increased in the SS salivary gland tissues and serum in vivo [91,93,94].

As such, chronic IFN activation and BAFF secretion by the innate immune and epithelial cells can promote the activation and survival of B cells independent of T cell help in SS (Figure 2).

## 7. Role of Inflammasome in SS

The inflammasome signaling pathway is another important pathway in innate immune response. Inflammasomes are multimeric protein complexes that mediate immune response upon the recognition of PAMPs and DAMPs. Inflammasome activity produces inflammatory cytokines IL-1β and IL-18. The structural components of inflammasomes include a PRR that recognizes PAMPs and DAMPs, pro-caspase-1, and adapter proteins such as apoptosis-associated speck-like protein containing a caspase recruitment domain (ASC), which connects the PRR and pro-caspase-1. PRR undergoes self-polymerization only upon the recognition of PAMPs, DAMPs, or intracellular changes, thereby forming the inflammasome complex from its structural components [95,96]. Among all inflammasomes, NOD-like receptor family pyrin domain-containing 3 (NLRP3) inflammasome activity is associated with SS pathogenesis (Figure 3).

The structural components of the NLRP3 inflammasome (NLRP3, ASC, and caspase-1) had significantly higher gene expression levels in the salivary glands in patients with SS. The gene expression levels of NLRP3 inflammasome structural components in the salivary glands was also positively correlated with lymphocytic salivary gland inflammation and autoantibody production in SS. In the salivary gland tissues, the mRNA expression of *P2X7R*, a receptor involved in the NLRP3 inflammasome activity, was found to be positively associated with IgG, antinuclear antibody, and Ro/SS-A levels [97]. In a previous research, P2X7R knockout mice did not develop salivary gland inflammation, which was induced by administering a P2X7R agonist in wild-type C57BL/6 mice [98]. In addition, in an autoimmune exocrinopathy mouse model of SS, P2X7R inhibition reduced inflammasome activity in the salivary glands, thereby increasing saliva secretion and eliciting a protective effect against salivary gland inflammation [99]. Dry eye was also caused by NLRP3 inflammasome signaling in a mice model [100].

The IL-1β and IL-18 levels in PBMCs increased in patients with active SS who have a high EULAR Sjӧgren’s Syndrome Disease Activity Index (ESSDAI) score. ESSDAI is calculated by physicians as the sum of activities multiplied by the weighted number of each region of glandular and extra-glandular lesions [101,102]. The expression levels of inflammasome structural components are high in patients who developed mucosa-associated lymphoid tissue lymphoma associated with SS progression [103]. Based on the above-mentioned information, inflammasome activity may promote SS progression, as evidenced by worsening disease severity and progression to lymphoma.

## 8. Conclusions

This study provided an overview of the role of innate immune response activation in SS. Chronic inflammation and tissue damage can be eliminated by suppressing the signaling pathway activating innate immune responses.

To date, medications targeting the TLR or TLR-dependent signaling pathways have been effective in the mouse models of several autoimmune diseases, and they are currently tested in humans [104,105,106]. As a therapeutic strategy focused on type I IFNs, anifrolumab, a monoclonal antibody that inhibits type I IFNs, has been found to be effective in SLE [107]. As mentioned previously, JAK/STAT inhibition was also effective in the murine models of SS and suppressed the activation of SGECs in SS in vitro [69,108]. Indeed, a clinical trial of JAK inhibitor is in progress in patients with SS [109].

Thus, local and systemic inflammation in SS can be suppressed by treatment strategies targeting the TLR signaling cascade and type I IFN activation.

Several trials have assessed the efficacy of hydroxychloroquine, which inhibits the activation of TLR3, TLR7, and TLR9 in SS. However, its efficiency remains controversial [110,111]. SS has several clinical features and is extremely heterogeneous, and not all patients exhibit type I IFN activity. Hence, the condition may require an individualized treatment strategy targeting signaling activity. Although PRRs can recognize different ligands including viral and autoimmune, the triggers of PRR involvement in SS pathogenesis remain unclear. Further, the mechanisms causing downstream signaling overactivity have not been elucidated. Thus, an assessment of the mechanisms involved in signal modulation and activation can lead to the development of novel and effective treatment strategies.

## Figures and Tables

**Figure 1 ijms-22-03090-f001:**
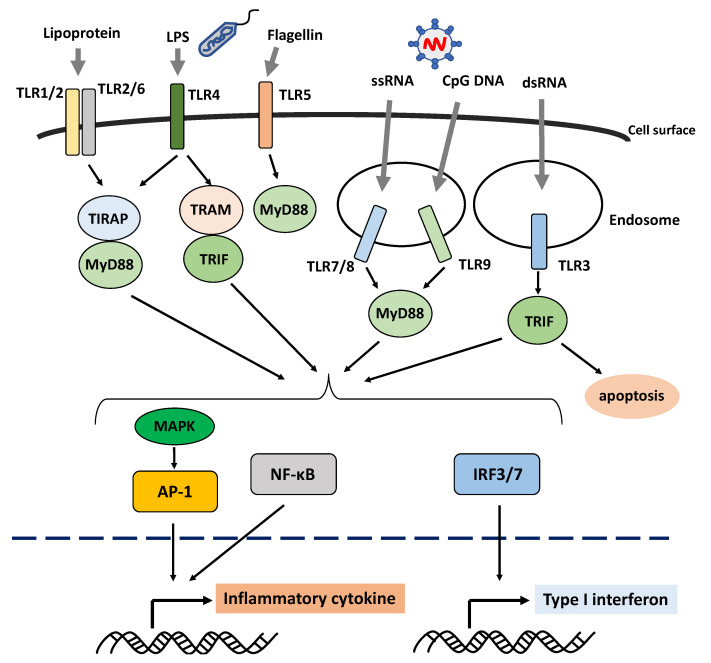
Schematic overview of the activation of the Toll-like receptor signaling pathway. Toll-like receptors (TLRs) are the pattern-recognition receptors, which recognize different pathogen- or danger-associated molecular patterns. TLR1, TLR2, TLR4, TLR5, and TLR6, which are involved in the recognition of proteins and fats, are mainly localized on the cell surface, whereas TLR3, TLR7, TLR8, and TLR9, which are involved in the recognition of nucleic acids, are mainly localized within the cells in the endosomes and endoplasmic reticulum. The TLR recognition of their ligands leads to the production of inflammatory cytokines and type I interferon via downstream molecules. Abbreviations: AP-1: activator protein 1; IRF: interferon regulatory factor; LPS: lipopolysaccharide; MAPK: mitogen-activated protein kinase; MyD88: myeloid differentiation primary response 88; NF-κB: nuclear factor kappa B; TIRAP: toll/interleukin-1 receptor domain-containing adapter protein; TLR: Toll-like receptor; TRAM: TRIF-related adaptor molecule; TRIF: TIRAP inducing interferon β.

**Figure 2 ijms-22-03090-f002:**
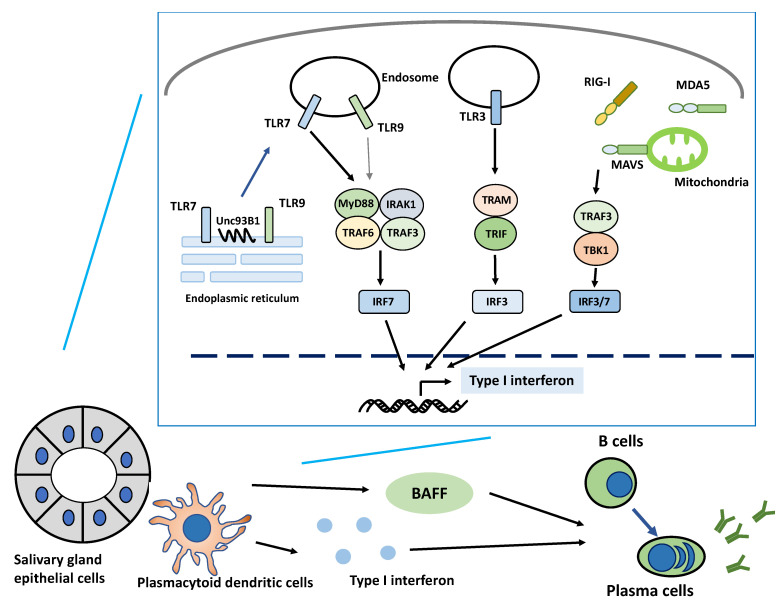
Activation of type I interferon production via endosomal Toll-like receptors and RNA sensor signaling pathway in Sjögren’s syndrome. Endosomal Toll-like receptors (TLRs), RNA sensor retinoic acid-inducible gene I (RIG-I) and melanoma differentiation-associated gene-5 (MDA5) expressed in salivary gland epithelial and infiltrating cells such as plasmacytoid dendritic and B cells in Sjögren’s syndrome induce mainly the production of type I interferon (IFN) via downstream molecules. The balance between TLR7 and TLR9 responses is controlled by Unc93B1, a protein responsible for trafficking TLR7 and TLR9 from the endoplasmic reticulum to the endosome, where TLR7 and TLR9 recognize nucleic acids. Disruption of this control may activate TLR7 signaling and lead to the overproduction of inflammatory cytokines and type I IFN. TLR7 ligand stimulation also enhances the expression of RIG-I and MDA5, synergistically producing a type I IFN signature. The overproduction of type I IFN specifically induces polyclonal B-cell expansion and B-cell differentiation into autoantibody-producing plasma cells. In addition, type I IFN-induced signaling can induce B-cell activating factor (BAFF) production and promote the activation and survival of B cells. The salivary gland epithelial cells in Sjögren’s syndrome are one of the source cells that produce BAFF after the stimulation of type I IFN. Abbreviations: BAFF: B cell-activating factor; IRAK1: interleukin-1 receptor-associated kinase 1; IRF: interferon regulatory factor; MAVS: mitochondrial antiviral signaling protein; MDA5: melanoma differentiation-associated gene-5; MyD88: myeloid differentiation primary response 88; RIG-I: RNA sensors retinoic acid-inducible gene I; TBK: TANK-binding kinase; TLR: Toll-like receptor; TRAF: tumor necrosis factor receptor-associated factor; TRAM: TRIF-related adaptor molecule; TRIF: TIRAP inducing interferon β.

**Figure 3 ijms-22-03090-f003:**
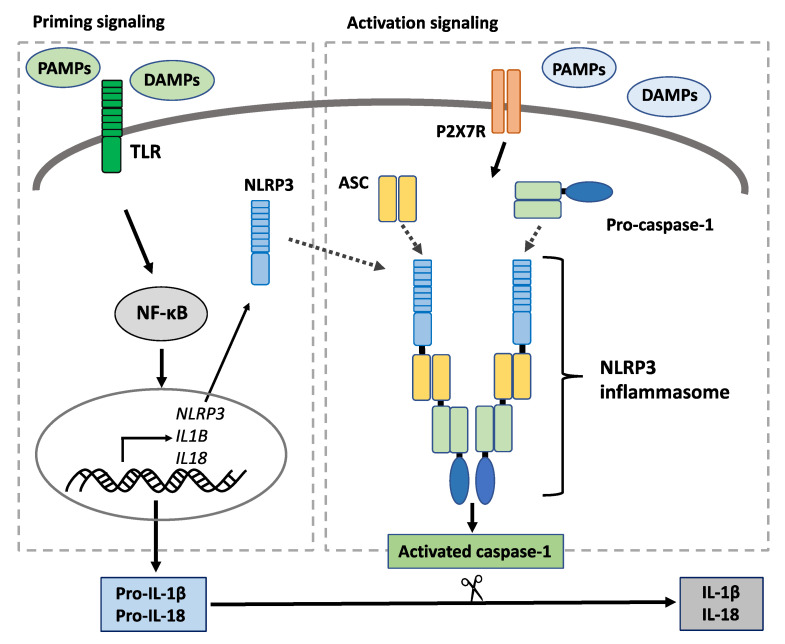
Mechanism of the NLRP3 inflammasome pathway. NLRP3 inflammasomes are formed by priming and activation signaling. In the priming signaling, several pathogen-associated molecular patterns (PAMPs) or danger-associated molecular patterns (DAMPs) bind to the pattern-recognition receptors (PRRs) such as Toll-like receptors and activate the NF-κB signaling pathway, which leads to the upregulation of NLRP3, pro-interleukin (IL)-1β, and pro-IL-18. In the activation signaling, various PAMPs, DAMPs, or intracellular changes induce the formation of the NLRP3 inflammasome composed of NLRP3 as a PRR, pro-caspase-1, and adapter proteins such as the apoptosis-associated speck-like protein containing a caspase recruitment domain that connects NLRP3 and pro-caspase-1. Then, the inflammasome complex activates pro-caspase-1. The activated caspase-1 cleaves the pro-IL-1β and pro-IL-18 into IL-1β and IL-18, which are biologically active forms. Abbreviations: ASC: apoptosis-associated speck-like protein containing a caspase recruitment domain; DAMP: danger-associated molecular pattern; IL: interleukin; NF-κB: nuclear factor kappa B; NLRP3: NOD-like receptor family pyrin domain containing 3; TLR: Toll-like receptor.

## Data Availability

Not applicable.

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
