# Peer review of "Role of the Innate Immunity Signaling Pathway in the Pathogenesis of Sjögren’s Syndrome"

_ijms, 2021, doi:10.3390/ijms22063090_

Round 1
Reviewer 1 Report
Manuscript ID: ijms-1146511
Type of manuscript: Review
Title: Role of the innate immunity signaling pathway in the pathogenesis of Sjögren’s syndrome
Toshimasa Shimizu, Hideki Nakamura, Atsushi Kawakami
The authors draw attention to the importance of the innate immune signal pathway in the development of inflammation and therefore, they illustrate the dysregulation of immune system in SS. The paper is well organized and well written and represents an important contribution to the field. I personally recommend the publication of this work.
I have only a suggestion: the authors could be to add some information on the principal mechanisms of inflammation connected to the trigger innate immune system
Author Response
We greatly appreciate your suggestions and comments, which have helped us improve our manuscript significantly. We have provided our responses to your comments.
Reply:
Thank you for your comment.Because we have described the activation of MAPK pathway, which one of the downstream signaling of TLRs in SS, we have added the sentence and figure with regard to the MAPK pathway in the ‘2. Innate immune response’ section and figure 1. In addition, because there are several reports on the involvement of the JAK/STAT pathway, which is an inflammatory cascade leading from type I IFN signaling, in the pathogenesis of SS, we have added the following sentence.
Page6, lines 18-24
‘Indeed, the expression of STAT3, JAK1, and JAK2 increased in the salivary glands of patients with SS [40,68]. Moreover, the inhibition of JAK suppressed the expression of IFN-related genes in SS-derived cultured SGECs. JAK inhibitor-treated NOD mice also showed increased salivary flow rates and reduced lymphocytic infiltration in salivary glands [69]. These findings suggested that the JAK/STAT pathway activated upon type I IFN signaling could contribute to SS pathogenesis.’
Page10, lines 21-23
‘As mentioned previously, JAK/STAT inhibitions were also effective in the murine models of SS and suppressed the activation of SGECs in SS in vitro [69,108]. Indeed, a clinical trial of JAK inhibitor is in progress in patients with SS [109].’
Moreover, we have changed the reference numbers.
Thank you again for your comments on our paper. We trust that the revised manuscript is suitable for publication.
Reviewer 2 Report
This review assessed the role of innate immune signal pathway in the development of inflammation and immune abnormalities in Sjogren's syndrome. An assessment of the mechanisms involved in signal modulation and activation can lead to the development of novel and effective treatment strategies. I would still like a review of more recent studies as almost two-thirds of the cited literature sources are older than 2016.Author Response
We greatly appreciate your suggestions and comments, which have helped us improve our manuscript significantly. We have provided our responses to your comments.
Reply:
Thank you for your comment. We have carefully reviewed recent research results with regard to SS and innate immunity and cited these related papers in this manuscript. However, these papers that describe the results of innate immunity pathway in pathogenesis of SS were concentrated before 2016. In addition, it was necessary to include older papers on the general theories of innate immunity and those of SS. In this revised manuscript, we have also added several recent findings on the JAK-STAT pathway activated upon type I IFN signaling in SS, and the potential therapeutic strategies of the JAK-STAT pathway as follows:
Page6, lines 18-24
‘Indeed, the expression of STAT3, JAK1, and JAK2 increased in the salivary glands of patients with SS [40,68]. Moreover, the inhibition of JAK suppressed the expression of IFN-related genes in SS-derived cultured SGECs. JAK inhibitor-treated NOD mice also showed increased salivary flow rates and reduced lymphocytic infiltration in salivary glands [69]. These findings suggested that the JAK/STAT pathway activated upon type I IFN signaling could contribute to SS pathogenesis.’
Page10, lines 21-23
‘As mentioned previously, JAK/STAT inhibitions were also effective in the murine models of SS and suppressed the activation of SGECs in SS in vitro [69,108]. Indeed, a clinical trial of JAK inhibitor is in progress in patients with SS [109].’
From the above, we have cited recent papers with regard to the JAK-STAT pathway and changed the referencenumbers.
Thank you again for your comments on our paper. We trust that the revised manuscript is suitable for publication.